# Seroprevalence of Pertussis in Adults at Childbearing Age Pre- and Post- COVID-19 in Beijing, China

**DOI:** 10.3390/vaccines10060872

**Published:** 2022-05-29

**Authors:** Zhiyun Chen, Jie Pang, Yuxiao Zhang, Yiwei Ding, Ning Chen, Nan Zhang, Qiushui He

**Affiliations:** 1Department of Medical Microbiology, Capital Medical University, Beijing 100069, China; chenzy@ccmu.edu.cn (Z.C.); 122019000048@ccmu.edu.cn (J.P.); chenning@ccmu.edu.cn (N.C.); nan@ccmu.edu.cn (N.Z.); 2Medical Research & Laboratory Diagnostic Center, Jinan Central Hospital Affiliated to Shandong First Medical University, Jinan 250013, China; zhangyuxiao0423@yeah.net; 3Department of Respiratory Medicine, The Sixth Medical Center of PLA General Hospital, Beijing 100048, China; dingyiwei@ccmu.edu.cn; 4Institute of Biomedicine, Research Center for Infections and Immunity, University of Turku, 20520 Turku, Finland

**Keywords:** pertussis, anti-PT IgG antibodies, seroprevalence, adults, China

## Abstract

The number of reported pertussis cases has significantly decreased during the coronavirus disease 2019 (COVID-19) pandemic under the influence of strict public health measures in many countries including China. This study evaluated the prevalence of serum anti-pertussis toxin (anti-PT) IgG antibodies in adults at childbearing age pre- and post- COVID-19 in Beijing, China. Altogether, 2021 serum samples collected from individuals aged 20 to 39 years who attended an annual health examination at the Sixth Medical Center of PLA General Hospital, Beijing, in 2018~2020 were measured by ELISA. The median concentration of anti-PT IgG antibodies among participants in 2020 (2.96 IU/mL) was significantly lower than that in 2018 (3.27 IU/mL) (*p =* 0.011) and in 2019 (3.24 IU/mL) (*p =* 0.014). The percentage of participants with anti-PT IgG antibodies higher than 40 IU/mL (indicating a pertussis infection within the past few years) was 1.79% (9/503) in 2018, 2.04% (15/735) in 2019 and 1.66% (13/783) in 2020, respectively. The corresponding numbers of the non-detectable (<5 IU/mL) rate of anti-PT IgG antibodies were 66.60%, 65.99% and 70.24%. Our results showed that there was a significant difference between true and reported incidence rates even during the COVID-19 pandemic. The proportion of adults at childbearing age without pertussis-specific antibodies is high, suggesting that booster vaccinations in adults should be considered in this country.

## 1. Introduction

Severe acute respiratory syndrome coronavirus 2 (SARS-CoV-2) causes coronavirus disease 2019 (COVID-19). During the pandemic, public health measures such as lockdowns, wearing facemasks and social distancing have been implemented in many countries to reduce the spread of COVID-19. These different measures have been found to be associated with significant reductions in several respiratory infections including pertussis [1,2,3,4,5,6,7,8].

Severe COVID-19 cases are rare among newborns and infants. However, serious consequences caused by other respiratory infections, such as pertussis, are more common in infants [9]. Several recent reports showed that the number of reported pertussis cases has significantly decreased during the COVID-19 pandemic [1,2,7,8]. In China, after the outbreak of COVID-19 in Wuhan in December 2019, strict public health measures were implemented in various provinces and cities. For example, in Beijing, since 9 February, 2020, the end of the extended Spring Festival holiday, the government has recommended remote working at home and online teaching at all school education systems to avoid personnel aggregation. Wearing facemasks was required most of the time when going out. The number of reported pertussis cases sharply decreased from 30,027 in 2019 to 4475 in 2020 in China. In contrast, the incidence of pertussis has continued to rise in recent years, including before the COVID-19 pandemic (Figure 1) [10].

In China, pertussis vaccination was introduced in the early 1960s. Diphtheria–tetanus–acellular pertussis (DTaP) vaccine has been in use since 2007, and completely replaced the diphtheria–tetanus–whole pertussis (DTwP) vaccine by 2013. Since 1982, infants have been administered three primary doses of DTwP or DTaP vaccines at the age of 3, 4, and 5 months, with a booster dose given at 18~24 months. DTP3 coverage has been over 99% in recent years (Figure 1), and reported incidence rates were less than 1.6/100,000 according to official data by 2018 [11,12]. However, the true incidence of pertussis is significantly underestimated in China, as shown in many studies conducted in different regions [13,14,15,16,17,18,19].

Beijing is the capital municipality directly under the central government and the national central city of the People’s Republic of China, having 16 districts and more than 21 million permanent residents. Haidian District, located in the west and northwest of the main urban area of Beijing, has a resident population of more than 3.1 million. The pertussis vaccination program for used in Beijing is the same as the national program mentioned above. According to the Beijing Public Health Information Center in 2020, pertussis vaccination coverage among the resident child population for the whole city was 99.94%, while in the Haidian District it was at 100% [20]. Similar to the whole nation, the number of reported pertussis cases in Beijing has been increasing, from less than 15 per year before 2013 to 188 in 2018 (Appendix A
Figure A1). The incidence rate was less than 0.1/100,000 before 2013 and increased to 0.4~0.9/100,000 in 2014~2018 (Appendix A
Figure A2) [12]. Moreover, a recent cross-sectional study conducted in Beijing compared the rate of undetectable anti-pertussis toxin (anti-PT) IgG antibodies (<5 IU/mL) in adults aged 20 to 39 years during two study periods (2010 and 2015/2016) and found that the rate of undetectable anti-PT IgG antibodies had significantly increased between the two periods, showing that the adult population were becoming vulnerable to pertussis [19]. That finding might partly explain the increased incidence of pertussis in the following years in this country. It is well known that parents and other family members are the main sources of infection for infants and young children [21]. Although the impact of the COVID-19 pandemic on reported pertussis cases has been reported, little is known about the seroepidemiology of pertussis among adults at childbearing age after the COVID-19 pandemic.

In this study, we aimed to evaluate the levels of serum anti-PT IgG antibodies in adults aged 20~39 years in 2018~2020 in Beijing and to compare the seroepidemiology of pertussis pre- and post- COVID-19 in these age groups. The study will provide important information on the impact of the COVID-19 pandemic on seroprevalence of specific anti-pertussis antibodies as well as the effectiveness of the current immunization strategy in China.

## 2. Materials and Methods

### 2.1. Study Subjects and Serum Samples

All serum samples were leftover sera from adults of childbearing age (20~39 years old) who attended annual health examinations in 2018~2020, and were collected at the Sixth Medical Center of PLA General Hospital in the Haidian District, Beijing, China. These individuals, who attended the annual heath examination, were residents of the Haidian District. The heath examination center of the hospital recorded the basic sociodemographic information of participants. The only information collected from the study subjects was age, gender, and date of sampling. The information on the vaccination status of these subjects was not available. According to the immunization program in China, all adults included in this study were born during 1980~1998 and many should have received DTwP vaccines during childhood. All sera were kept at −80 °C before analysis.

Altogether, 1165 serum samples were collected from June to August in 2018, 1631 were collected from January to December in 2019, and 1314 were collected from August to November in 2020. It should be emphasized that since the outbreak of COVID-19 in December 2019, cities in China have implemented lockdown management, and a majority of people stayed at home for remote working and studying until August, 2020. Of these serum samples, 503 (43.2%) in 2018, 735 (45.1%) in 2019 and 783 (59.6%) in 2020 were included in the study. The samples of each year were grouped by age or sex, and the demographic data of the study subjects are shown in Table 1.

### 2.2. Serological Testing

Commercial ELISA kits (Institut Virion/Serion GmbH, Würzburg, Germany) were used to determine concentration of anti-PT IgG antibodies according to the manufacturer’s instructions. The interpretation of the results was described previously [10]. Antibodies greater than or equal to 100 IU/mL indicated a recent infection within one year, and a value between 40 and 100 IU/mL indicated an infection within the past few years. When the concentration values were below 5 IU/mL, it was considered that no antibodies were detected. When the concentration was between 5 and 40 IU/mL, it was considered seronegative.

### 2.3. Statistical Analysis

Data were analyzed using GraphPad Prism version 7 (San Diego, CA, USA) and SPSS version 25.0 (SPSS Inc., Chicago, IL, USA). Serum anti-PT IgG concentrations in different age groups and genders were analyzed by a normality test. Normally distributed continuous variables of two or more groups were compared using a Student’s *t*-test or one-way ANOVA. Non-normally distributed continuous variables of two or more groups were compared using the Mann–Whitney U test or Kruskal–Wallis test. The mean or median concentration of each group was compared with one of every other group when there were more than two groups, and *p* value was adjusted to account for multiple comparisons. Seroprevalence and the proportion of subjects with non-detectable anti-PT IgG in adults between different years were compared by the Chi-square test. Two-tailed *p* values < 0.05 were considered statistically significant.

## 3. Results

A total of 2021 serum samples were included in the study. Of them, 503 collected in 2018 included 276 males and 227 females, 735 collected in 2019 included 407 males and 328 females, and 783 collected in 2020 included 385 males and 398 females (Table 1).

The median concentration of anti-PT IgG antibodies among all subjects in 2020 (2.96 IU/mL) was significantly lower than that in 2018 (3.27 IU/mL) (*p =* 0.011) and in 2019 (3.24 IU/mL) (*p =* 0.014). Nine (1.79%), 15 (2.04%) and 13 (1.66%) subjects had anti-PT IgG antibodies higher than 40 IU/mL in 2018, 2019 and 2020, respectively. No significant differences in seroprevalence were found among the three years. Among the nine subjects whose anti-PT IgG antibodies were higher than 40 IU/mL in 2018, two (0.40%), including one man and one woman, had anti-PT IgG antibodies higher than 100 IU/mL. One (0.14%) male subject in 2019 and one (0.13%) male subject in 2020 had anti-PT IgG antibodies higher than 100 IU/mL. No significant differences in study subjects with anti-PT IgG antibodies higher than 100 IU/mL were observed among the three years (Table 1).

In 2018, the median concentration of anti-PT IgG antibodies among adults aged 20~29 was found to be higher than that among those aged 30~39 (3.67 IU/mL vs. 2.94 IU/mL, *p =* 0.002), whereas no such differences were found between these two age groups in 2019 and 2020. In contrast, there was no difference in seroprevalence between the 20~29 y and 30~39 y age groups in any year of the study period (Table 1).

In 2020, the median concentration of anti-PT IgG antibodies in men was higher than that in women (3.26 IU/mL vs. 2.78 IU/mL, *p =* 0.018). However, no such difference was found between different genders in 2018 and 2019 (Table 1). The seroprevalence in men in 2020 was found to be higher than that in women (2.86% vs. 0.50%, *p =* 0.011), while no statistical differences were found between men and women in 2018 and 2019 (Table 1).

The non-detectable (less than 5 IU/mL) rate of anti-PT IgG antibodies was 66.60%, 65.99% and 70.24% in 2018, 2019 and 2020, respectively (Figure 2). No differences were found when they were compared with each other. Among adults aged 20~29 in 2018, anti-PT IgG antibodies were not detected in 71.31% (179/251) of tested sera, which was higher than that among adults aged 30~39 (61.90%, 179/251) (*p =* 0.025). No such difference was found between the age groups of 20~29 years and 30~39 years in 2019 and 2020, and between genders in any year of the study period (Figure 3).

## 4. Discussion

In the era of pre- COVID-19, pertussis has been one of the significant public health problems throughout the world. Despite high childhood vaccination coverage, epidemic peaks of pertussis have occurred every 2 to 5 years during the past decades [22,23,24]. Pertussis is no longer just a childhood disease. Outbreaks and seroepidemiological data observed in many countries suggest that vaccine-acquired immunity has waned in adolescents and adults, resulting in a large number of unprotected populations. Typical clinical characteristics of pertussis include a paroxysmal cough and whooping. However, in adults, pertussis is usually less severe and presents atypical symptoms. This change in clinical manifestations means that pertussis can often be unrecognized and is largely under-diagnosed in adult populations. Meanwhile, it poses a transmission risk to infants who are too young to have completed the primary vaccinations [25,26]. As with many countries, the incidence of pertussis in China is most likely under-reported [13,19].

Because the pertussis vaccination policy has remained unchanged in China since the 1980s and no booster vaccination is used in children after the age of 2 years, in this study, those adults who were of childbearing age and had anti-PT IgG antibodies ≥40 IU/mL were considered to have a real *B. pertussis* infection. In our previous study, we found that the seroprevalence of anti-PT IgG antibodies in adults aged 20~39 years was 5.1% in 2010 and 4.0% in 2015/2016 [19]. However, in this present study, the corresponding seroprevalence of anti-PT IgG antibodies was about 1.66%~2.04% in 2018~2020. No statistical difference has been found between the pre- and post- COVID-19 eras, but the median concentration of serum anti-PT IgG antibodies in 2020 was lower than those in 2018 and 2019. The difference was probably because Beijing was not completely locked down from February 2020, and the collection time of serum samples included in this study was from August to November 2020, during which the lockdown had ended in this city. Although the positivity rates were decreased, they were still about 100-fold higher than the reported incidence rates in corresponding years reported in Beijing and the whole of China (Figure 1, Appendix A
Figure A2) [12].

In contrast to the epidemic cycle of pertussis observed in many countries, no obvious epidemic peaks have been noticed in China since 2007, when DTaP was implemented nationwide. Multiple reasons could explain the difference, e.g., variations in clinician awareness of pertussis, laboratory diagnostic methods used, and surveillance and reporting systems among different provinces and cities. In addition, no booster vaccinations are used after two years of age, resulting in a large susceptible population in this country that could not be excluded [19]. However, the reported pertussis cases have been increasing in recent years, especially from 2017 to 2019, the pre- COVID-19 era (Figure 1). In 2020, because of COVID-19 and its intervention measures, the number of reported cases sharply decreased, to 4475, only about one sixth of the number of cases reported in 2019 (Figure 1) [10]. The reduction in the incidence of pertussis reported in China was in agreement with the incidence rates reported in many other countries [2,7,8]. Interestingly, the seroprevalence of pertussis observed among adults at childbearing age in 2020 did not seem to be significantly different in 2018 and 2019.

The reason for the correlation between seroprevalence of pertussis and gender was not clear in this study. The positivity rates in men with anti-PT IgG antibodies higher than 40 IU/mL tended to be higher than those in women during the study period, but the statistical difference between genders was found only in 2020. However, there was no significant difference in positivity rate in men with anti-PT IgG antibodies higher than 100 IU/mL indicative of a recent infection within a year. A similar result was also found in our previous study conducted in adults of the same age in 2010 [13]. Although some studies have reported higher seropositivity in male subjects [27,28], similar seropositivity rates in men and women were also reported [29,30]. Compared to the age group of 20~29 years, a higher rate of undetectable anti-PT IgG antibodies in adults aged 30~39 years was observed in 2018. However, similar results were not found either in other periods in this study or in other studies [30].

Although the intervention measures used during the COVID-19 pandemic have reduced the number of reported pertussis cases, our seroprevalence study indicated that true incidence is most likely underestimated. It should be kept in mind that during the COVID-19 pandemic, all of the diagnostic efforts for respiratory tract infections were focused on COVID-19. Until now, in China, the differential diagnosis of COVID-19 from other respiratory tract infections including pertussis has not been considered, and there are no specific analytical approaches or kits for simultaneous specialized diagnosis of pertussis during the diagnosis of COVID-19. Therefore, pertussis cases were probably going undiagnosed. Over the past decade, *B. pertussis* isolates not expressing the vaccine antigens PRN, PT or FHA have appeared. It is worth noting that the emergence of strains that do not express PT is a challenge for serological diagnosis relying solely on anti-PT IgG antibodies, which may also have caused some deviations in our results. However, so far, no more than 10 PT negative strains have been reported [31,32]. On the other hand, the increase in the proportion of people without specific anti-PT IgG antibodies also suggests the risk of declined herd immunity. In our previous study, we found the rate of undetectable anti-PT IgG antibodies in adults who were of childbearing age had significantly increased from 29.1% in 2010 to 57.4% in 2015/2016 [19]. In this study, we found that the undetectable rate had been further increased to 65.99%~70.24% in 2018~2020, suggesting that the proportion of adults of reproductive age susceptible to pertussis has been increasing. It is well known that vaccination is the most effective tool to prevent pertussis. Since pertussis continuously circulates in adults, a booster dose in this age group including maternal immunization is recommended in many industrial countries [25,26]. However, so far, no booster immunizations in China have been recommended.

There are some limitations in this study. First, the vaccination status of the study subjects was not available. However, adults included in this study were older than 20 years, and they should have not received any booster vaccination because no booster dose is given after two years of age in China. Therefore, the anti-PT antibodies detected were most likely due to infection rather than vaccination. Secondly, the serum samples included in this study were collected from subjects who attended annual health examinations. The majority of people were in a healthy state, without any acute clinical manifestations including cough when undergoing the examination. However, medical history of diagnosed pertussis infection was not known. Thirdly, the sociodemographic data of these study subjects were deficient.

## 5. Conclusions

Our results showed that, in China, although the number of reported pertussis cases significantly decreased after the COVID-19 pandemic in 2020, the seroprevalence of anti-PT IgG antibodies in adults of childbearing age was close to that observed in the pre- COVID-19 era, suggesting that the true incidence of pertussis is still significantly underestimated in China. Therefore, it is extremely important to maintain high vaccination coverage during the pandemic. Further, the proportion of adults who did not have pertussis-specific antibodies has been increasing, even up to 70% since the COVID-19 pandemic, suggesting that a large number of adults of childbearing age are vulnerable to pertussis. To protect against pertussis in infants and adults, booster immunizations in adults should be considered in China.

## Figures and Tables

**Figure 1 vaccines-10-00872-f001:**
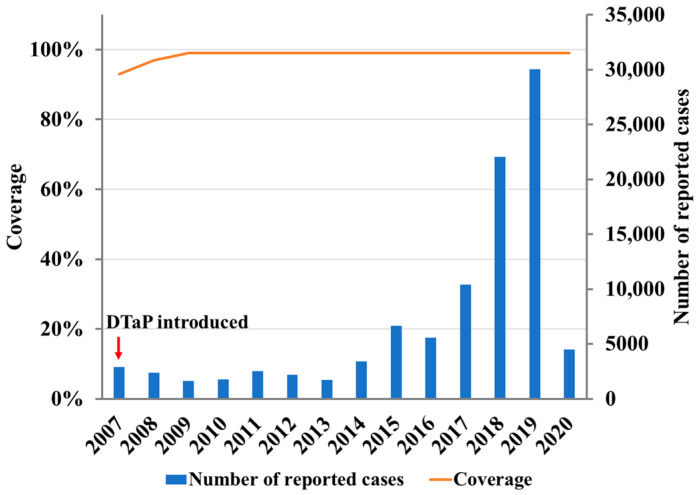
DTP3 vaccination coverage and number of reported pertussis cases in China, 2007–2020 [10,11]. Adapted with permission from Ref. [10]. 2022. Geneva: World Health Organization (WHO). Licence: CC BY-NC-SA 3.0 IGO. DTP3: diphtheria tetanus toxoid and pertussis (DTP)-containing vaccine, 3rd dose.

**Figure 2 vaccines-10-00872-f002:**
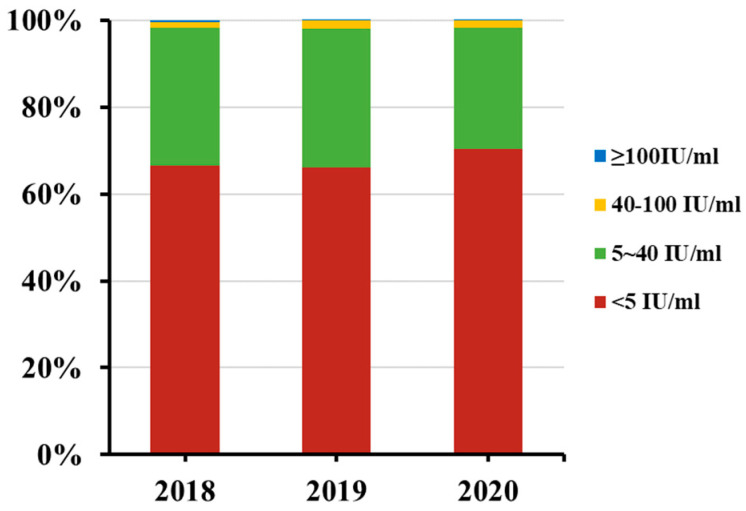
Distribution of serum anti-PT IgG antibody concentrations in adults aged 20~39, Beijing, China, 2018–2020. The numbers of serum specimens with anti-PT IgG antibody concentrations ≥100 IU/mL, 40~100 IU/mL, 5~40 IU/mL, and <5 IU/mL in each group were calculated, and the data are shown as percentages. No statistical difference was observed among these three years.

**Figure 3 vaccines-10-00872-f003:**
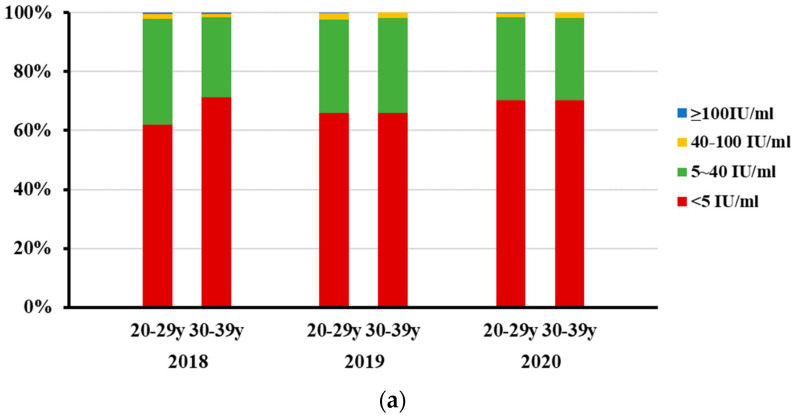
Distribution of serum anti-PT IgG antibody concentrations in adults aged 20~39, Beijing, China, 2018–2020. The numbers of serum specimens with anti-PT IgG antibody concentrations ≥100 IU/mL, 40~100 IU/mL, 5~40 IU/mL, and <5 IU/mL in each group were calculated, and the data are shown as percentages. (**a**) Distribution of serum anti-PT IgG antibody concentrations in adults aged 20~29 y and 30~39 y in each year. A significant difference was found in the non-detectable (<5 IU/mL) rate of anti-PT IgG antibodies between the 20~29 y and 30~39 y age groups in 2018 (*p* = 0.025). (**b**) Distribution of serum anti-PT IgG antibody concentrations in males and females in each year. No statistical differences were observed in non-detectable (<5 IU/mL) rate of anti-PT IgG antibodies between genders in any year.

**Table 1 vaccines-10-00872-t001:** The concentration and positivity rate of anti-PT IgG antibodies between genders in adults aged 20~39 years, Beijing, China, 2018~2020. (n = 2021) ^1^.

Year	Group	n	GMC (95% CI) (IU/mL)	Median (95% CI) (IU/mL)	≥40 IU/mL		≥100 IU/mL	
					n	Positive (%)	n	Positive (%)
2018	Total	503	3.82(3.52–4.14)	3.27(2.99–3.71)	9	1.79	2	0.40
	Age							
	20~29 y	252	4.28(3.82–4.81)	3.67(3.14–4.24)	5	1.98	1	0.40
	30~39 y	251	3.40(3.04–3.80)	2.94(2.50–3.50) ^4^	4	1.59	1	0.40
	Sex							
	Male	276	3.61(3.24–4.03)	3.08(2.73–3.41)	6	2.17	1	0.36
	Female	227	4.08(3.62–4.60)	3.78(2.99–4.32)	3	1.32	1	0.44
2019	Total	735	3.81(3.55–4.09)	3.24(3.01–3.51)	15	2.04	1	0.14
	Age							
	20~29 y	327	3.78(3.38–4.22)	3.12(2.81–3.51)	8	2.45	1	0.31
	30~39 y	408	3.84(3.51–4.21)	3.35(3.01–3.81)	7	1.72	0	0.00
	Sex							
	Male	407	3.84(3.49–4.23)	3.27(2.95–3.64)	10	2.46	1	0.25
	Female	328	3.78(3.40–4.19)	3.18(2.88–3.76)	5	1.52	0	0.00
2020	Total	783	3.34(3.13–3.56)	2.96(2.72–3.31) ^2^^,3^	13	1.66	1	0.13
	Age							
	20~29 y	347	3.31(3.00–3.65)	2.97(2.64–3.39)	5	1.44	1	0.29
	30~39 y	436	3.63(3.09–3.66)	2.96(2.59–3.45)	8	1.83	0	0.00
	Sex							
	Male	385	3.65(3.33–4.01)	3.26(2.79–3.80)	11	2.86	1	0.26
	Female	398	3.06(2.81–3.33)	2.78(2.43–3.10) ^5^	2	0.50 ^6^	0	0.00

^1^ A cut-off ≥100 IU/mL indicates a recent infection and that ≥40 IU/mL indicates a recent infection in the last few years. ^2^ A significant difference was found in median concentration of anti-PT IgG antibodies among all subjects between 2018 and 2020 (*p =* 0.011). ^3^ A significant difference was found in median concentration of anti-PT IgG antibodies among all subjects between 2019 and 2020 (*p =* 0.014). ^4^ A significant difference was found in median concentration of anti-PT IgG antibodies between subjects aged 20~29 y and 30~39 y in 2018 (*p =* 0.002). ^5^ A significant difference was found in median concentration of anti-PT IgG antibodies between men and women in 2020 (*p* = 0.018). ^6^ A significant difference was found in positivity rate between men and women in 2020 (*p =* 0.011). CI: confidence interval.

## Data Availability

Reported pertussis cases and DTP3 coverage in China during 2007–2020 were summarized and visualized according to data from the WHO Immunization data pool at https://immunizationdata.who.int/compare.html?COMPARISON=type1__WIISE/MT_AD_COV_LONG+type2__WIISE/MT_AD_INC_LONG+option1__DTP_coverage+option2__PERTUSSIS_cases&CODE=CHN&YEAR= (accessed on 3 April 2022), and the National Health Commission of the People’s Republic of China at http://www.nhc.gov.cn/jkj/s7923/202108/7337fd75c8b749309a2de28aec1a03bd.shtml (accessed on April 2022). Reported pertussis incidence rates in China during 2007–2018 were summarized and visualized according to data from the Data-center of China Public Health Science at https://www.phsciencedata.cn/Share/ky_sjml.jsp?id=09163f05-0e42-4e24-a524-dc6ddf8206ac (accessed on April 2022). Other data presented in this study are available on request from the corresponding author. The data are not publicly available due to intellectual property considerations.

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
