# Peer review of "Seroprevalence of Pertussis in Adults at Childbearing Age Pre- and Post- COVID-19 in Beijing, China"

_vaccines, 2022, doi:10.3390/vaccines10060872_

Round 1

Reviewer 1 Report

In this study, the authors investigated the serum anti-PT IgG antibodies in adults at a childbearing age (20~39 years old during 2018~2020) in Beijing for pertussis seroepidemiology comparison pre- and post- COVID-19. Also, meaningful information about the COVID-19 pandemic’s impact on the seroprevalence of specific anti-pertussis antibodies and the success of the ongoing immunization strategy in China has been reported. This manuscript is within the journal’s scope. It is well written and interestingly addressed. I recommend a major revision of the paper before publication in this journal. Therefore, the manuscript eventually can be published after some improvements and corrections:

  1. The abbreviation should be specified once in the first related place and then used repeatedly.
  2. English needs to be improved. Some sentences were incomprehensible.
  3. The conclusion should be improved, and statistical data should be considered.
  4. In the discussion part, discuss the requirement of booster vaccination in case of pertussis antigen changes and hence its different diagnosis in some individuals.
  5. In the discussion part, please explain the following phrase and debate its mechanism: “public health measures used during COVID-19 pandemic may partly interfere with the circulation of B. pertussis in the community and cause reduced exposure to the pathogen, therefore resulting in the further increase in the proportion of adults who are vulnerable to pertussis.”
  6. Please describe the specific analytical approaches associated with COVID-19 respiratory tract infections diagnosis for specialized diagnosis of pertussis.
  7. How many times were the measurements repeated to avoid errors?
  8. What is the control group in this study?

Reviewer 2 Report

The authors are to be thanked for their considerable efforts in performing this extensive study. This cross-sectional study had  very important results, however to my understand it is important to define the following aspects:

1) The history of pertussis vaccination must be describe in the introduction, We are not clear about the date an type of vaccination in china.

2) It must be clarifying to describe participants' pertussis vaccination history by using the national card of immunization, or self-report immunization.

3) Cellular and Accelular vaccines could have some different results in the interpretation of anti-PT IgG antibodies in adults from 20 to 29 or 30 to 39 years old.

4) Inclusion and exclusion criteria should be detailed, and sociodemographic characteristics and incidence rate in the geographic zone must be described, using national incidence is not enough to make any extrapolation of the data.

5) According to the discussion authors suggest that booster in adults  is needed as a part of the national immunization program, but a R0=10 needs more than 90% of vaccine coverage in order to prevent an outbreak even during the COVID pandemic. Authors must describe if there is a possible selection bias and how it will address the results and conclusions.

5)

Reviewer 3 Report

the article is a seroprevalence study on Ab pertussins in childbearing population in China.

As presented by the Authors seroprevalence studies on Ab pertussis are largely published, also in China. Could you state what is the originality of this study? What does it add to the literature? 

sociodemographic data are very poor (only age and sex)

vaccination status is a piece of fundamental information and it was not collected by the authors. could you explain why?

many elements of the strobe checklist are missing. Please consider revising the manuscript accordingly (for instance: description of the setting, description of the variables, methods used to collect the information, and so on).

conclusions: could you add some public health implications of your results?

Round 2

Reviewer 1 Report

Now, the manuscript can be accepted without any further changes.

Reviewer 3 Report

I recognized the efforts performed by the authors in order to meet my suggestions.

However, I suggest considering the paucity of sociodemographic data and the missing data on vaccination status in the limitations section. 
